# Effect of Imposed Shear Strain on Steel Ring Surfaces during Milling in High-Speed Disintegrator

**DOI:** 10.3390/ma13102234

**Published:** 2020-05-13

**Authors:** Karel Dvořák, Adéla Macháčková, Simona Ravaszová, Dominik Gazdič

**Affiliations:** 1Faculty of Civil Engineering, Brno University of Technology, Veveří 331/95, 602 00 Brno, Czech Republic; ravaszova.s@fce.vutbr.cz (S.R.); gazdic.d@fce.vutbr.cz (D.G.); 2Faculty of Materials Science and Technology, VŠB–Technical University of Ostrava, 17. listopadu 15, 708 00 Ostrava, Czech Republic; adela.machackova@vsb.cz

**Keywords:** disintegrator, microscopy, wear, high energy milling, cement

## Abstract

This contribution characterizes the performance of a DESI 11 high-speed disintegrator working on the principle of a pin mill with two opposite counter-rotating rotors. As the ground material, batches of Portland cement featuring 6–7 Mohs scale hardness and containing relatively hard and abrasive compounds with the specific surface areas ranging from 200 to 500 m^2^/kg, with the step of 50 m^2^/kg, were used. The character of the ground particles was assessed via scanning electron microscopy and measurement of the absolute/relative increase in their specific surface areas. Detailed characterization of the rotors was performed via recording the thermal imprints, evaluating their wear by 3D optical microscopy, and measuring rotor weight loss after the grinding of constant amounts of cement. The results showed that coarse particles are ground by impacting the front faces of the pins, while finer particles are primarily milled via mutual collisions. Therefore, the coarse particles cause higher abrasion and wear on the rotor pins; after the milling of 20 kg of the 200 m^2^/kg cement sample, the wear of the rotor reached up to 5% of its original mass and the pins were severely damaged.

## 1. Introduction

The necessity to enhance the utility properties of materials and increase the longevity of construction components has led research and development in various industrial branches, such as materials processing [1] or construction [2,3], towards the design of new materials and improving the performance of contemporary ones. Generally, enhancement of the mechanical and utility properties can be achieved via variations in the chemical composition of the particular material, as documented, for example, by Zhang et al. [4], and structural modifications (grain refinement), which was demonstrated also for pure and commercially pure metals [5,6]. Verlinden et al. claimed that grain refinement introduces lattice defects, such as dislocations and grain boundaries, which increase the materials’ intrinsic energy and act as strengthening agents [7]. Kunčická et al. studied syntheses of various kinds of alloys and reported that the grain size can typically be decreased in two ways, the first of which is application of the the methods of intensive/severe plastic deformation (SPD) decreasing the grain size down to sub-micron scale via imparting severe shear strain [8] (e.g., equal channel angular pressing [9] and high-pressure torsion [10]). The second is the implementation of powder metallurgy, i.e., the production of materials from original powders, which is typically used for challenging materials, such as Ti composites [11] or tungsten heavy alloys [12]. Combinations consisting of the processing of challenging powder-based materials, such as Ti-based [13], and W-based [14] alloys, via SPD methods are also advantageous.

The process of production of a component from powders involves final shape and surface treatments, which typically follow deformation processing and/or variations of sintering and heat treatments. The first production step is, however, preparation of the original powders. These are usually manufactured by milling. The grain size distribution and morphology of the grains have fundamental effects not only on the sintering process itself, as mentioned by Wang et al. [15], but also on the final structure of the alloys as described by Macháčková et al. [16]. Surzhenkov et al. studied the wear resistance and mechanisms of abrasion during milling. Based on the results of their research, it can be stated that the grain refinement in mills is typically ensured via several fundamental mechanisms, among which are compression, shear (attrition), compression pulse, impact (stroke or collision), impact and shear, tension/bending, splitting, and cutting. Milling can be performed by collisions of particles with the working tools, via mutual collisions between particles, or between particles and their environment [17].

Among the recent trends in milling is the high-energy milling (HEM) method. The possibilities of application of HEM are in various industrial fields, from milling of demanding composites in metallurgy, as documented, for example, by Muroi et al. [18] or Sazavi et al. [19], through fabrication of various intermetallic and ceramic materials, such as those mentioned by Serena et al. [20], to the preparation of nanomaterials (e.g., Rojac et al. [21]). High-speed grinding (HSG) is a particular type of HEM carried out by applying high amounts of energy using very short and intense power pulses. The amount of energy that is effectively transferred to the material is higher in HSG than in the case of conventional grinding in mills with identical power inputs. Among the types of mills suitable for HSG is a high-speed pin mill with two counter-rotating rotors, known as the disintegrator. This type of mill, the material in which is refined by high-frequency changes in mechanical strength, was extensively studied by Hint [22]. The disintegrator is particularly suitable for the grinding and activation of fine powder materials, as documented by Bumanis [23], and Bumanis and Bajare [24]. The principle of a disintegrator lies in accelerating the material to a high speed by means of the pins on the grinding rotor. The particles than collide with other particles or with the pins on the rotor which rotates in opposite direction. Disintegrators only use the impact, impact and shear, shear (attrition), and tension/bending mechanisms. However, other processes similar to those in attritors or jet mills occur as well. These are introduced by turbulent flow and rapid compression and expansion between the rotors, as described by Kovalev [25]. Baláž wrote in his study [26] that the main advantages of a high-speed disintegrator are continuity of the grinding process, and variety of working tools that can be employed. On the other hand, Surzhenkov et al. stated the main disadvantage to be that the grinding elements are prone to abrasion/wear [17].

Collisions of particles and hard surfaces can be advantageous since the shear strain introduced to the hard surface via the impacting particles typically induces the formation of hard structure phases at the surface of the base material (such as martensite for stainless steels as reported by Staman et al. [27]), by the effect of which the surface hardness increases, and the surface hardening consequently reduces the tendency of the base material (i.e., rotors) to abrasion/wear, as claimed by Silva et al. [28] and Han et al. [29]. On the other hand, continuous high-speed shelling of the base material (rotors) with hard and/or sharp-edged particles inevitably leads to abrasion/wear in time. Gåhlin and Jacobson [30] mentioned that the intensity of wear depends on the mineralogy, morphology, and granulometry of the ground material. However, the effect of particle size of the milled material on wear is not yet fully understood. Misra and Finnie [31] suggested that increasing the particle size increases the wear rate. Small particles lead to penetrations that do not pass through the surface layers of the grinding elements; according to the theory introduced by Larsen-Bads [32], sufficiently small particles are in elastic contact with the grinding tools and do not contribute to abrasion. The presupposed effect of particles’ shapes on the degree of abrasion is based on a theory assuming that blunt shapes exert less pressure on the grinding tools and consequently act more like particles providing surface hardening via imposing shear strain and produce less wear than sharp-edged particles. For milled material featuring wider size fractions, the largest grains exert strong point loads on the surfaces of the grinding elements; i.e., such material is more abrasive [17,30].

If the pins are damaged due to abrasion, the efficiency of grinding is reduced. By this reason, the geometry of the milling elements is favourable to be kept constant for as long as possible during milling. The presented study is focused on characterization of the behaviour of rotors of a high-speed disintegrator during milling of a Portland cement, because the HSG technology seems to be a very effective milling technology, moreover, with the benefit of mechanical activation. However, certain issues in this field still remain uncovered. The standard Portland cement is relatively fragile and contains four main minerals with the average hardness of 6–7 (according to the Mohs scale). It features sharp-edged particles, which makes it a very abrasive material. Grinding by a disintegrator is also supposed to be very effective for this particular material since it responds very well in milling by impact or compression pulse [33]. The study is supplemented with characterization of correlations between entry granulometry and abrasion of the rotors during milling.

## 2. Materials and Methods

The equipment investigated is the DESI 11 disintegrator (Desintegraator Tootmise OÜ, Tallinn, Estonian Republic), which is a laboratory version of a high-speed pin mill with two counter-rotating rotors. The total installed output of the mill is 4.1 kW. The rotor rotation frequency is up to 12,000 RPM, and the maximum circumferential speed of each rotor is 92.4 m/s. The material is fed into the device by a continuous feeder and enters the grinding chamber through the middle of the left rotor; a schematic depiction of the principle of the disintegrator is shown in Figure 1a, and also described in [17]. The real machine is presented in Figure 1b. The construction of the mill allows for a choice of working tools. For this experiment, the CR type rotors designed and manufactured by the FF servis s.r.o. company (Prag, Czech Republic) were used. The rotors were manufactured from C45 steel, sometimes also characterized as SI 1045 steel. This medium-carbon steel was selected for its high quality, relatively high strength, and easy machinability; the rotors were machined from normalized hot-rolled bars. The external diameter of both the rotors was 147 mm. The left rotor has two rows, while the right rotor has three rows of 3 × 3 × 3 mm cubic pins. The designs of both the rotors are demonstrated in Figure 1c,d.

The Portland cement used for this experiment was prepared in a ball mill by collective milling of cement clinker (from Hranice cement plant) and chemical gypsum (Pregips), the chemical compositions of which are depicted in Table 1. The chemical composition of the selected clinker is typical for Portland clinkers. The gypsum (highly pure) contained relatively high humidity and it was further dried before milling to decrease the moisture to under 5%. The ratio of clinker to gypsum was 95:5.

For the subsequent milling process, a set of seven individual batches with specific surface areas ranging from 200 to 500 m^2^/kg was prepared. The batches of cement with the selected granulometries were fed into the disintegrator continuously. The dosing rate was 5.5 g·s^−1^ and the cement particles are supposed to spend approximately 1 second in the milling chamber due to the pin mill principle. All the samples were milled at the maximum speed of 12,000 RPM.

Grinding of all the samples was carried out under standard laboratory conditions, at 22 °C and relative humidity of 56%. After grinding of each 1 kg batch, thermal imprint on the rotor was acquired using a Flir E4 thermal camera (FLIR Systems Inc., Wilsonville, OR, USA). Between milling of the individual batches, the disintegrator was cooled by an air stream to the initial temperature of 22 °C and the rotors were replaced with a new set.

The increase in the Blaine specific surface area and the morphology of grains were determined for all the samples. The Blaine specific surface area was measured using a PC-Blaine-Star automatic device (Zünderwerke Ernst Brün GmbH, Haltern am See, Germany) with a measuring cell with the volume of 7.95 cm^3^. Measurement was repeated three times and averaged to minimize errors. To evaluate the impact of the input granulometry on the abrasion of working elements, two 20 kg batches of cement were prepared in a ball mill. The first sample featured the specific surface area of 200 m^2^/kg, while for the second one it was 450 m^2^/kg. The milling parameters were identical as for the previous case; rotor speed of 12,000 RPM and dosing rate of 5.5 g·s^−1^.

The particle size and particle size distribution were determined for both the input samples using a Malvern Mastersizer 2000 (Malvern Panalytical B.V., Almelo, The Netherlands) with a Hydro 2000 G fluid dispersing unit, 2-isopropanol was used as a dispersing agent. The morphology and grain shapes were observed and assessed by scanning electron microscopy (SEM) using a Tescan MIRA 3XMU (Tescan Brno s.r.o., Brno, Czech Republic) equipment.

Evaluation of the abrasion-milling elements was performed via calculating the ratio between the weight of the eroded rotors after milling of 1, 10 and 20 kg of abradant, and the initial weight of the rotors. The accuracy of the weight loss measurements was 0.01 mg.

Also, the impact of rotor wear on grinding efficiency was measured by monitoring the increase in the specific surface area of the cement. The abrasion of the rotors was also determined by 3D scanning. Detailed 3D scans of the rotor pins were performed and evaluated using an Olympus DSX1000 digital microscope (Olympus Czech Group, s.r.o., Prague, Czech Republic).

## 3. Results

Abrasion speed is not constant and depends on the input granulometry and on the shape of the ground particles. Morphology of the cement particles was investigated via SEM-BSE; images of the cement particles within the individual samples are depicted in Figure 2a–f within which individual pairs of figures represent morphology of the samples before and after milling in the HSG mill. Figure 2a,d depict the samples with the initial specific surface area of 200 m^2^/kg, whereas Figure 2b,e show the samples with the initial specific surface area of 300 m^2^/kg, and Figure 2c,f depict the samples with the initial specific surface area of 450 m^2^/kg.

The particles from the sample with the specific surface area of 200 m^2^/kg exhibited sharp edges, however, they were slightly abraded. Similar results were acquired for the sample with the specific surface area of 300 m^2^/kg, but these particles exhibited a more substantial abrasion. The particles from the finest fractions then exhibited more or less spherical shapes and the tendency to agglomerate. Agglomerated clusters were also present in the sample with the initial specific surface area of 450 m^2^/kg.

Laser granulometry revealed that the batch with the input-specific surface area of 200 m^2^/kg contained 20.8% of particles larger than 100 µm, while the batch with the input specific surface area of 450 m^2^/kg contained only 10.7% of particles larger than 100 µm. Aggregates were most probably not present because the measurements were performed under wet conditions.

The resulting specific surface areas for all the seven ground samples, together with the absolute and relative values of their increases, are depicted in Figure 3. The results show that the absolute specific surface area values increased significantly when grinding coarse fractions (specific surface areas of 200 and 250 m^2^/kg). In contrast, both the absolute and relative specific surface area values were lower for the input fractions of 300 and 350 m^2^/kg. The absolute increase in the specific surface area then increases gradually with continuing refinement of the input material, whereas the relative increase in the specific surface area remains more or less constant from the 300 m^2^/kg input fraction.

Thermal imprints on the rotors before and after milling of 1 kg batches of 200 m^2^/kg to 500 m^2^/kg samples are shown in Figure 4a–i, respectively. The imprints clearly show that grinding of the coarse batches (200 and 250 m^2^/kg) was primarily performed by the front faces of the rotor pins, the temperature of which increased (Figure 4c,d, respectively). As the specific surface area of the input cement increased, the areas in the vicinity of the pins started to exhibit increased temperature, too. The 300, 350, and 400 m^2^/kg batches contained plenty of coarse particles, the grinding of which was performed directly on the rotor pins. However, mutual grinding of fine particles against each other in the spaces between the pins occurred as well. This phenomenon was then dominant for the batches with the specific surface areas of 450, and 500 m^2^/kg, during the milling of which the pins acted more like breaking wedges directing the milled material into the spaces between them. For these fine batches, the particles were primarily milled by their mutual collisions, and milling by direct impacts with the pins’ front faces was of low significance (Figure 4h,i clearly show the pins to be cooler than the surrounding area).

The impact of input granulometry on wear of the grinding rotors is demonstrated by Figure 5 showing wear of the rotors (expressed as loss of rotor mass in relation to the original rotor mass) is dependent on the amount of the ground material. The figure indicates that a coarser material damages the grinding rotors of the disintegrator more significantly than a finer one.

After milling of only a 10 kg batch of the testing cement with the specific surface area of 200 m^2^/kg, the wear of the grinding rotor with three rows of pins was 2.14%, and the grinding rotor with two rows of pins exhibited wear of 1.96%. The wear directly influenced the increase in the specific surface area of particles, as the grinding was about 1.61% less effective. Table 2 summarizes the results of measurements of the specific surface area for the cement powder at the beginning of the experiment, and after milling of 10 and 20 kg batches. Table 3 then presents the reductions of grinding efficiency, characterized as reduction of the specific surface area. When grinding the fine-grained input material with the specific surface area of 450 m^2^/kg, the abrasion occurred to a lesser extent. The weight loss measured for the rotor with three rows of pins after grinding of a 10 kg batch was 0.28%, while for the rotor with two rows of pins it was 0.13%; the reduction of the efficiency of grinding by the worn rotors was negligible. Milling of a 20 kg batch of the testing cement resulted in further rotor weight loss; after grinding an additional 10 kg batch, the weight loss of the three-row rotor was 0.42%, while for the two-row rotor it was only 0.30%. The efficiency of the rotors was only 1.28% lower when compared to new rotors.

Figure 6a–h show 3D optical microscopy images of the new and worn rotor pins. All the scans were performed on the rotors with three rows of pins. Figure 6a,b depict the 3D scan and height profile, respectively, of the pins of a new rotor, while Figure 6c,d depict the 3D scan and height profile, respectively, of the pins of the rotor used for milling of a 20 kg batch of the 450 m^2^/kg sample. Figure 6e,f then depict the 3D scan and height profile, respectively, of the pins of the rotor used for the milling of a 20 kg batch of the 200 m^2^/kg sample. The figures confirm the wear analyses, as they clearly show that the wear was the most significant for the rotor used for milling of the 200 m^2^/kg batch; scans of a pin from the most worn rotor and its coloured height profile are shown in Figure 6g,h. The figures also show that the wear of the pins was the most significant at their front faces, and also at the face on the side from which the cement particles were injected into the disintegrator. The figures also show evident “shade” behind the pins, which corresponds to the above described occurring milling mechanisms for the individual sample batches.

## 4. Discussion

The different acting particle grinding principles imparted by different input granulometries have considerable effects on the abrasion of the grinding rotors, especially the pins. When milling the samples with the input-specific surface areas of 200 and 250 m^2^/kg, the pins of the grinding rotors are strongly heated due to the direct contact with the material being ground, as confirmed by the thermal imprints shown in Figure 4a,b. The direct crushing and friction of the coarse particles on the front faces of the rotor pins also resulted in a substantial increase in the specific surface areas of the particles. The negative aerodynamic pressure occurring right behind the rotor pins is not sufficient to impart swirling motion to the solid particles. For this reason, the particles are not milled by mutual collisions and thus maintain their sharp-edged morphology during passing through the mill; only a slight abrasion of their edges occurs. The wear of the rotors was the highest for these batches and expressed as mass % of the original rotor mass was almost 5% for both the rotors with two and three rows of pins. The shapes of the rotor pins were also severely damaged after milling of the batches with coarse particles, as demonstrated in Figure 6e–h. These findings are in accordance with the theory of Misra and Finnie [31]; large particles impact the front faces of the pins, easily penetrating the surface layers, and cause rapid erosion.

The results acquired during grinding of the samples with very fine particles having the specific surface areas of over 400 m^2^/kg were different; the cement particles exhibited intensive refinement and gained spherical shapes, however, they also exhibited a tendency to agglomerate. The momentum of the particles is sufficiently low for them to be drawn into the area of lower pressure in the space behind the rotor pins; the pins act more like breaking wedges and regulate the material flow. The milling is primarily performed by mutual particle collisions in the turbulent area right behind the rotor pins, as demonstrated by the thermal imprints on the rotors in Figure 4e–g, the locations with the highest temperatures in which were situated behind the rotor pins. In contrast to the coarse samples, the finer material with the specific surface area of 450 m^2^/kg flows around the pins. Thus small particles do not penetrate the surface layer of the grinding pins and the wear on the rotors is significantly lower for the fine samples. The 3D profiles of the rotor pins depicted in Figure 6c,d exhibited only a slight abrasion, which is consistent with the measured wear being lower than 0.5% of the original rotor mass.

Characterization of the grinding process for the batches featuring the input specific surface areas of 300 to 350 m^2^/kg is quite complex. Increases in the specific surface areas for these samples were lower than for the other, coarser as well as finer, batches. The thermal imprints depicted in Figure 4c,d show that a portion of the ground material is in direct contact with the front faces of the grinding pins. Nevertheless, an increase in the temperature in the spaces between and behind the rotor pins is observed as well. This phenomenon implies that during milling of these samples, reduction of particles’ sizes proceeds not only via direct contact with the rotor pins, but also, to some extent, via mutual collisions of the particles behind the grinding pins. The amount of coarse grains that are ground on the front surfaces of the rotor pins is lower than for coarser samples. At the same time, the amount of fine grains that have lower momentum and are milled by mutual collisions is not sufficient. The total increase in the specific surface area is thus lower than for the finer and coarser samples. The sharp edges of the particles are abraded after passing through the mill, however, the shape of the coarse particles does not change significantly. For these reasons, assessment of the surface hardening effect for these particular batches would require a deeper study.

## 5. Conclusions

A variety of modern materials with enhanced properties is nowadays fabricated via methods of powder metallurgy, the initial powders for which are typically manufactured using milling machines. This study focused on the characterization of the performance of a DESI 11 high-speed disintegrator. For the purpose of the study, a variety of samples of Portland cement, containing relatively hard and abrasive particles with specific surface areas ranging from 200 to 500 m^2^/kg, were used. The results showed that:Grinding of coarse angular-shaped particles proceeds primarily via impacts of particles with the front faces of rotor pins, and such particles cause their significant damage and abrasion.Fine cement particles are refined in the area between and behind the pins. This grinding principal supports wear reduction and the grinding efficiency is not substantially affected when milling larger batches of finer cement samples.Without sustaining significant damage, this type of high-speed mill is suitable for final grinding of fine particles with the Blaine specific surface area of at least 400 m^2^/kg.Coarser materials can be ground more efficiently in other types of mill.A reasonable specific surface increase with minimum wear of the grinding tools can be achieved in a relatively short time for materials with suitable input granulometries.

The high-speed disintegrator is, thus, an advantageous, versatile piece of equipment for the preparation of powders subsequently used to prepare modern materials.

## Figures and Tables

**Figure 1 materials-13-02234-f001:**
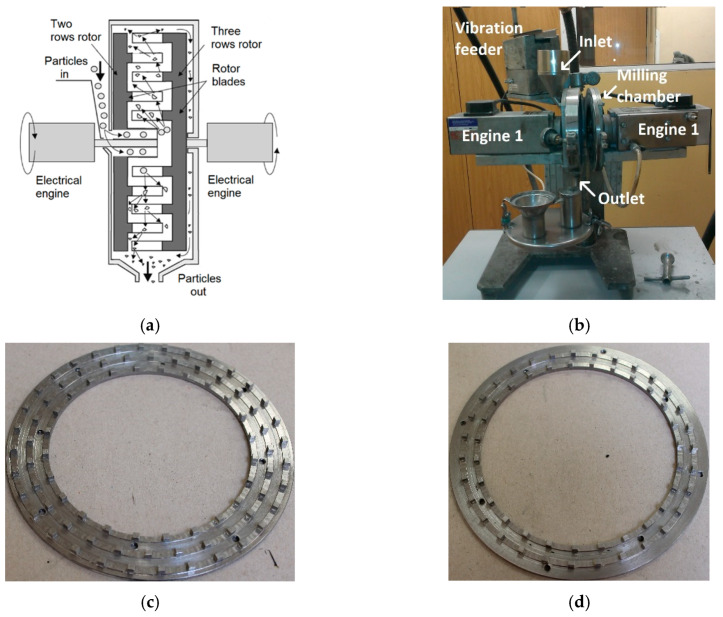
(**a**) Schematic depiction of the investigated disintegrator; (**b**) laboratory DESI 11 HSG mill (**c**) rotor with three rows of pins; (**d**) rotor with two rows of pins.

**Figure 2 materials-13-02234-f002:**
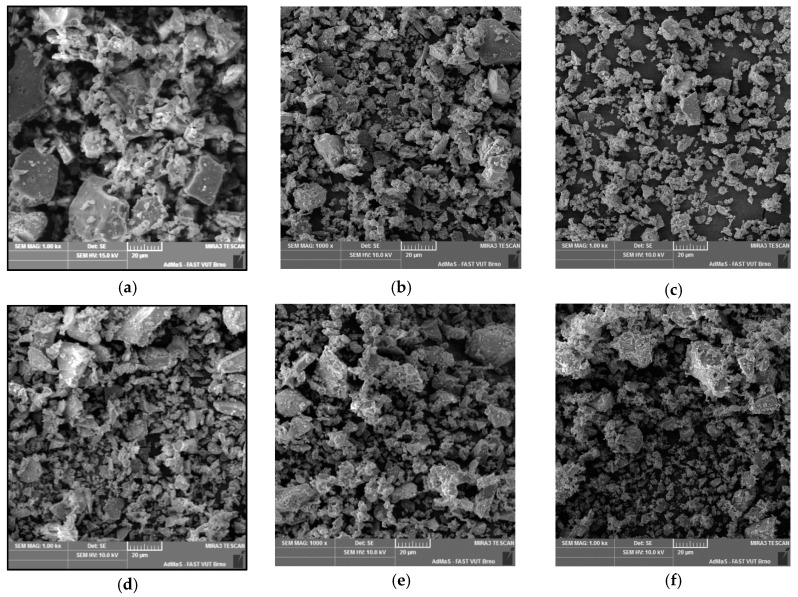
Comparison of the original and post-milling shapes of cement particles for: 200 m^2^/kg sample, (**a**) original particles; (**d**) milled particles; 300 m^2^/kg sample, (**b**) original particles; (**e**) milled particles; 450 m^2^/kg sample, (**c**) original particles; (**f**) milled particles.

**Figure 3 materials-13-02234-f003:**
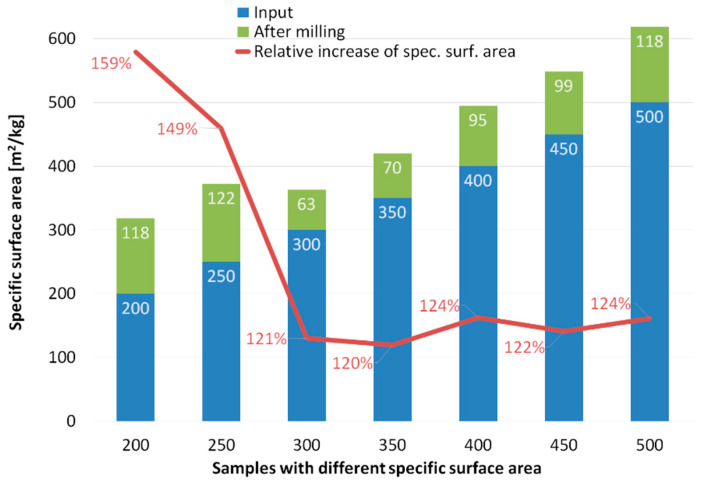
Increase in specific surface area for all ground samples.

**Figure 4 materials-13-02234-f004:**
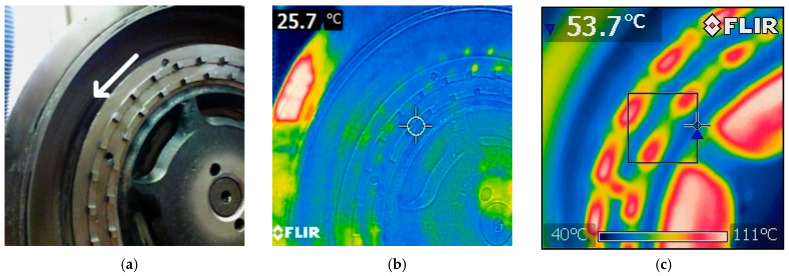
Thermal imprints on rotors before and after milling of 1 kg of individual batches: (**a**) new rotor with two rows of pins (rotation direction is marked by white arrow); (**b**) new rotor; (**c**) 200 m^2^/kg; (**d**) 250 m^2^/kg; (**e**) 300 m^2^/kg; (**f**) 350 m^2^/kg; (**g**) 400 m^2^/kg; (**h**) 450 m^2^/kg; (**i**) 500 m^2^/kg.

**Figure 5 materials-13-02234-f005:**
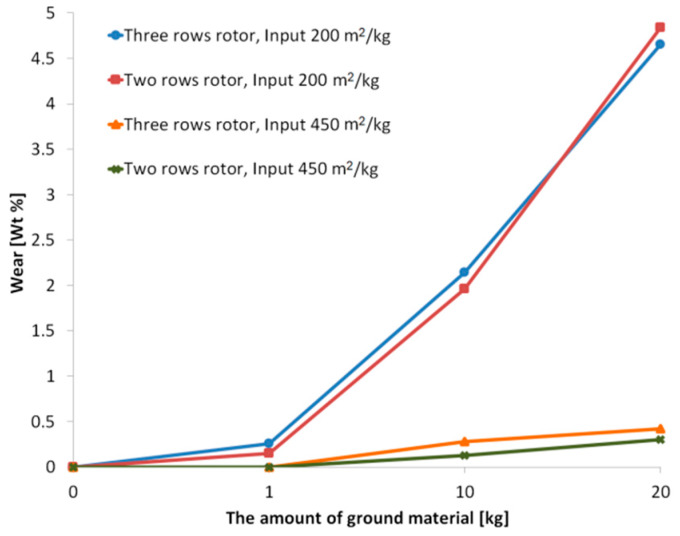
Effect of the amount of ground material on rotor wear for individual samples.

**Figure 6 materials-13-02234-f006:**
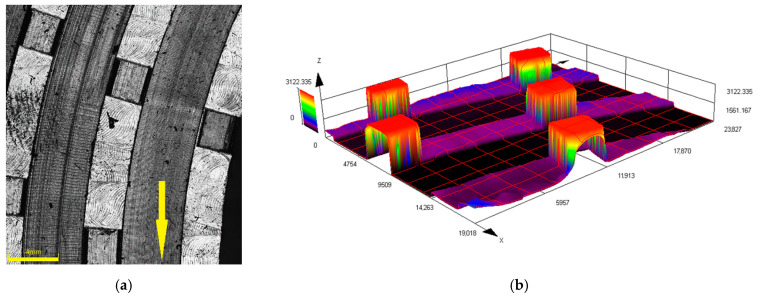
Optical images of new rotor: (**a**) 3D scan and (**b**) height profile of pins; optical images of rotor used for milling of 20 kg batch of 450 m^2^/kg sample: (**c**) 3D scan and (**d**) height profile of pins; optical images of rotor used for milling of 20 kg batch of 200 m^2^/kg sample: (**e**) 3D scan and (**f**) height profile of pins. Detailed images of pin marked by circle in (**f**) from rotor used for milling of 20 kg batch of 200 m^2^/kg sample: (**g**) detailed scan and (**h**) coloured profile. (Rotation directions marked by arrows).

**Table 1 materials-13-02234-t001:** Partial chemical compositions of the ground material.

Material	Component								
	SiO_2_	CaO	Al_2_O_3_	Fe_2_O_3_	SO_3_	CaSO_4_∙2H_2_O	H_2_O	CaSO_4_	Others
Clinker	20.29	65.33	5.21	5.04	0.79	-	-	-	3.34
Gypsum	-	-	-	-	-	84.0	11.0	2.4	2.6

**Table 2 materials-13-02234-t002:** Specific surface area of the cement powder at the beginning and after 10 and 20 kg.

Input Specific Surface Area (m^2^/kg)	Specific Surface Area of Resulting Powders (m^2^/kg)		
	1 kg	10 kg	20 kg
200	310	305	273
450	548	543	541

**Table 3 materials-13-02234-t003:** Reduction of grinding efficiency.

Input Specific Surface Area (m^2^/kg)	Grinding Efficiency Reduction (%)		
	1 kg	10 kg	20 kg
200	-	1.61	11.94
450	-	0.91	1.28

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
