# Peer review of "Effect of Imposed Shear Strain on Steel Ring Surfaces during Milling in High-Speed Disintegrator"

_materials, 2020, doi:10.3390/ma13102234_

Round 1
Reviewer 1 Report
The authors should give some additional information, as indicated by comments in the text.

Author Response
Thank you for the comments. All the changes are marked in the revised text by "Track Changes" function. My response to your comments is in the pdf file attached to this letter.

Reviewer 2 Report
The reviewer comments of the paper «Effect of imposed shear strain on steel rings surfaces during milling in high-speed disintegrator»
- Reviewer
The authors presented an article «Effect of imposed shear strain on steel rings surfaces during milling in high-speed disintegrator». Reviewed article is very interesting and write at good scientific level. Figures, tables as well as terminology are clear and precise. Analysis are detailed and well described, scientific arguments were used to define the potential of presented method. However, there are several points in the article that require further explanation.
Comment 1:
The whole introduction is well written. At the end, give the purpose of the article.
Comment 2:
Give the article a methodology for measuring the wear of rotors. What devices and equipment are used?
Comment 3:
Authors need to replace "," with "." in the designation of fractions in Fig. 3.
Comment 4:
Present some photos with rotors wear with a description.
Comment 5:
Conclusions should be improved. It is necessary to more clearly show the novelty of the article and the advantages of the proposed method. What is the difference from previous work in this area? Show practical relevance. It is necessary to give quantitative and qualitative indicators of the proposed method. What is the difference from other researchers. Conclusions should correspond to the purpose of Article.
Draw conclusions in the form of highlights:
- first conclusion
- second conclusion
- etc.
The topic of the article is interesting and fairly well written. After minor changes can an article be considered for publication in the "Materials".
Author Response
Thank you very much for your comments.
All changes are marked in the revised text by "Track Changes" function.
Comment 1:
The whole introduction is well written. At the end, give the purpose of the article.
Answer: ACCEPTED: The motivation was clarified at the end of the „introduction“ chapter
Comment 2:
Give the article a methodology for measuring the wear of rotors. What devices and equipment are used?
Answer: ACCEPTED Partially: The rotor abrasion was determined by the weight differences between initial and after milling weight. This is clarified in the text now.
Comment 3:
Authors need to replace "," with "." in the designation of fractions in Fig. 3.
Answer: ACCEPTED: The fig. 3 was changed.
Comment 4:
Present some photos with rotors wear with a description.
Answer: ACCEPTED Partially: Wear of the rotors´ pins are demonstrated in Fig. 6. Some explanation has been added. Better description was added into the paragraph below table 2 and 3.
Comment 5:
Conclusions should be improved. It is necessary to more clearly show the novelty of the article and the advantages of the proposed method. What is the difference from previous work in this area? Show practical relevance. It is necessary to give quantitative and qualitative indicators of the proposed method. What is the difference from other researchers. Conclusions should correspond to the purpose of Article.
Draw conclusions in the form of highlights:
first conclusion
second conclusion
Answer: ACCEPTED: Conclusion has been rewritten according to your suggestions.
Reviewer 3 Report
The paper analyzes a topic with practical applicability. The research is done rigorously.
I have the following observations:
Line 96 - The paper specifies that the rotors is made of C45. Please specify the standard of the material, its condition (possibly heat treatment, hardness…...)
Lines 143 ... 145 -Under figure 2, the information is not complete. Not all figures are fully explained.
153 - In figure 3 the information is not suggestively presented. Values that have different units of measurement are drawn on the same bar. Please redo the figure. For each experiment I ask you to show comparatively, graphically (by bars) the input specific surface area and nearby the final specific surface area ( can be both on the same bar). The relative increase in the specific surface area can be presented as a distinct curve.
212, 228, 251, 253 - References are made to possible changes of the rotors hardness without any hardness measurement being made. It is advisable not to make statements that are not supported by experimental research.
Author Response
Thank you very much for your comments.
Especially last comment was really useful for us. All changes are marked in the revised text by "Track Changes" function.
Line 96 - The paper specifies that the rotors is made of C45. Please specify the standard of the material, its condition (possibly heat treatment, hardness…...)
Answer ACCEPTED: Better description of the steel C45 was added to the „Materials and methods“ chapter.
Lines 143 ... 145 -Under figure 2, the information is not complete. Not all figures are fully explained.
Answer: ACCEPTED: Paragraph situated above the fig. 2 was modified and moved under the figure 2. Some additional information about agglomeration were added.
153 - In figure 3 the information is not suggestively presented. Values that have different units of measurement are drawn on the same bar. Please redo the figure. For each experiment I ask you to show comparatively, graphically (by bars) the input specific surface area and nearby the final specific surface area (can be both on the same bar). The relative increase in the specific surface area can be presented as a distinct curve.
Answer: ACCEPTED: Fig. 3 was changed according to the suggestions.
212, 228, 251, 253 - References are made to possible changes of the rotors hardness without any hardness measurement being made. It is advisable not to make statements that are not supported by experimental research.
Answer: ACCEPTED: Thank you for this comment. I agree with that and I erased these sentences. We are planning to measure the hardness changes. It could be part of the next study.
Round 2
Reviewer 2 Report
All comments are satisfied. The article may be published in its current form in "Materials".
Author Response
Thank you for all your comments.